## Theories

enzyme kinetics; Hessian matrix; invertase; Jacobian matrix; mathematical modelling.

**Author for correspondence:**
T. Nägele,
E-mail: thomas.naegele@lmu.de

# Metabolic regulation of subcellular sucrose cleavage inferred from quantitative analysis of metabolic functions

Thomas Nägele

Faculty of Biology, Plant Evolutionary Cell Biology, Ludwig-Maximilians-Universität München, Planegg-Martinsried, Germany

## Abstract

Quantitative analysis of experimental metabolic data is frequently challenged by non-intuitive, complex patterns which emerge from regulatory networks. The complex output of metabolic regulation can be summarised by metabolic functions which comprise information about dynamics of metabolite concentrations. In a system of ordinary differential equations, metabolic functions reflect the sum of biochemical reactions which affect a metabolite concentration, and their integration over time reveals metabolite concentrations. Further, derivatives of metabolic functions provide essential information about system dynamics and elasticities. Here, invertase-driven sucrose hydrolysis was simulated in kinetic models on a cellular and subcellular level. Both Jacobian and Hessian matrices of metabolic functions were derived for quantitative analysis of kinetic regulation of sucrose metabolism. Model simulations suggest that transport of sucrose into the vacuole represents a central regulatory element in plant metabolism during cold acclimation which preserves control of metabolic functions and limits feedback-inhibition of cytosolic invertases by elevated hexose concentrations.

## 1. Introduction

The quantitative study of biochemical reaction networks represents an interdisciplinary research area of (bio)chemistry, physics and mathematics. Enzymes catalyse chemical reactions under physiologically relevant conditions. Enzyme activity directly depends on temperature, pH, ion strength and redox potential of a cell or compartment showing characteristic optima (Arcus & Mulholland, 2020; Bisswanger, 2017). In addition, enzyme activity in cellular systems is affected and regulated by diverse biochemical effectors, for example, comprising other proteins and metabolites (Atkinson, 1969; Chen et al., 2021). As a result, cellular enzyme activity represents a variable of biochemical networks which is shaped by a large parameter space challenging experimental, but also theoretical, analysis. Enzyme kinetic models mathematically describe enzymatic reaction rates as a function of one or more parameters and variables. In general, biochemical kinetics is based on the mass action law assuming the reaction rate to be proportional to the probability of reactant collision (Waage & Gulberg, 1864; 1986). This probability is proportional (a) to the concentration of reactants, and (b) to the number of molecules of each reactant that participate in a reaction, that is, to the power of molecularity. The rate $v$ of a reaction following the mass action law with molecularities $m_i$ and $m_j$ of substrates $S_i$ and products $P_j$, respectively, is described by the rate equation (equation (1)):

$$v = v_f - v_b = k_f \prod_{i=1}^{l_i} S_i^{m_i} - k_b \prod_{j=1}^{l_j} P_j^{m_j}. \tag{1}$$

Here, $k_f$ and $k_b$ represent the rate constants, that is, proportionality factors, for the forward ($k_f$) and backward ($k_b$) reaction. Introducing the reversible formation of an enzyme-substrate complex (E + S → ES), a release of product P from ES (ES → E + P), and the simplifying assumption that formation of ES is much faster than its decomposition into E and P, finally

yields the Henri–Michaelis–Menten kinetics (Henri, 1902; Henri & Hermann, 1903; Michaelis & Menten, 1913). Due to its capability to accurately describe and quantify mechanisms of enzyme catalysis and regulation, the Michaelis–Menten equation is crucial for biochemical understanding (Cornish-Bowden, 2015). It was derived based on experimental observations of sucrose hydrolysis, catalysed by invertase enzymes (Brown, 1902; Michaelis & Menten, 1913). Within this reaction, the glycosidic bond of sucrose is hydrolysed, and glucose and fructose are released (equation (2)):

$$\xrightarrow{r_{\text{in}}} \text{Suc} \xrightarrow{r_{\text{inv}}} \text{Glc} + \text{Frc} \xrightarrow[r_{\text{out,Frc}}]{r_{\text{out,Glc}}}. \tag{2}$$

In this kinetic model, $r_{\text{in}}$ represents the rate of sucrose biosynthesis, $r_{\text{inv}}$ the rate of invertase-driven hydrolysis and $r_{\text{out,Glc}}$ and $r_{\text{out,Frc}}$ hexose consuming processes, for example, phosphorylation by hexokinase enzymes. The corresponding ODE model of this reaction system describes sucrose, fructose and glucose dynamics by the sum of in- and effluxes (equations (3)–(5)):

$$\frac{d}{dt}\text{Suc} = r_{\text{in}} - r_{\text{inv}} = f(\text{Suc}), \tag{3}$$

$$\frac{d}{dt}\text{Glc} = r_{\text{inv}} - r_{\text{out,Glc}} = f(\text{Glc}), \tag{4}$$

$$\frac{d}{dt}\text{Frc} = r_{\text{inv}} - r_{\text{out,Frc}} = f(\text{Frc}). \tag{5}$$

The right side of the ODEs, that is, the sum of reactions, is summarised by metabolic functions $f$ and their integration yields the time course of metabolite concentrations. Dynamics of substrate and product concentrations can then be described by differential equations (DEs). If concentration dynamics are considered (only) over time, ordinary differential equations (ODEs) are applied while partial DEs account for more than one independent variable, for example, time and space.

In the following paragraph, invertase-catalysed sucrose hydrolysis is quantitatively explained and analysed down to a subcellular level applying an enzyme kinetic model based on Michaelis–Menten enzyme kinetics and experimental data of previous studies (further details about mathematical analysis are provided in the Supplementary Material). Invertases play a central role in diverse processes of plant metabolism, development and response to environmental stress (Koch, 2004; Ruan, 2014; Vu et al., 2020; Weiszmann et al., 2018; Xiang et al., 2011). Plant invertases occur in different isoforms with different compartmental localisation and biochemical properties (Sturm, 1996; Tymowska-Lalanne & Kreis, 1998). Both plant vacuolar and extracellular invertases possess an acidic pH optimum between 4.5 and 5.0 while cytosolic invertase has a neutral pH optimum between 7.0 and 7.8 (Sturm, 1999). Acidic and neutral invertases hydrolyse sucrose with a $K_{\text{M}}$ in a low-millimolar range (Sturm, 1999; Unger et al., 1992). Invertases are product inhibited, with glucose acting as a non-competitive inhibitor and fructose as a competitive inhibitor (Sturm, 1999). While biochemistry and kinetics of plant invertase reactions have been analysed in numerous studies, the physiological role of different subcellular isoforms and their regulatory impact on plant stress and acclimation reactions remain elusive. Further, due to its participation in cyclic sucrose breakdown and re-synthesis, the experimental study of invertase reactions remains challenging, particularly under changing environmental

**Table 1.** Carbon uptake rates and kinetic parameters of invertase-catalysed sucrose hydrolysis in *Arabidopsis thaliana,* accession Col-0, at 22 and 4°C.

| Kinetic parameter | Absolute value | Dimension |
| --- | --- | --- |
| $r_{\text{in,22°C}}$ | 8 | (µmol Suc h$^{-1}$ gFW$^{-1}$) |
| $r_{\text{in,4°C}}$ | 0.75 | (µmol Suc h$^{-1}$ gFW$^{-1}$) |
| $V_{\text{max,inv,22°C}}$ | 35.2 | (µmol Suc h$^{-1}$ gFW$^{-1}$) |
| $V_{\text{max,inv,4°C}}$ | 2 | (µmol Suc h$^{-1}$ gFW$^{-1}$) |
| $K_{\text{M,Suc}}$ | 12 | (µmol Suc gFW$^{-1}$) |
| $K_{i,\text{Frc}}$ | 0.23 | (µmol Frc gFW$^{-1}$) |
| $K_{i,\text{Glc}}$ | 0.12 | (µmol Glc gFW$^{-1}$) |

*Data source*: Kitashova et al. (2021).

conditions. Due to such cycling structures, it remains difficult to estimate metabolite amounts, their dynamics and effects on other segments of metabolic networks (Reznik & Segrè, 2010). Previous work has suggested a dominant role of invertase-driven sucrose cycling in regulation and stabilisation of primary metabolism and photosynthesis (Geigenberger & Stitt, 1991; Weiszmann et al., 2018). Here, metabolic functions of sucrose and hexoses are quantified to analyse compartment-specific invertase reactions in context of subcellular metabolite transport during plant cold exposure to evaluate its impact on metabolic acclimation.

## 2. Results and discussion

Due to the regulatory plasticity of metabolism, metabolite concentrations may vary significantly under similar environmental conditions and without stress exposure. For example, sucrose and hexoses may accumulate significantly, and even double in amount, during the light period of a diurnal cycle (Brauner et al., 2014; Seydel et al., 2022; Sulpice et al., 2014). Such strong dynamics of reaction product and substrate concentrations aggravate the quantitative analysis of metabolic regulation due to their non-linear impact on enzymatic rates. It follows that instead of analysing one (single) snapshot, a broad range of physiologically relevant metabolite concentrations and/or enzyme parameters needs to be analysed in order to cope with metabolic plasticity. Here, an example of such an analysis is provided applying a kinetic parameter set of invertase reactions (Table 1), which has previously been determined in *Arabidopsis thaliana* under ambient (22°C) and low (4°C) temperature (Kitashova et al., 2021).

Reaction rates of invertase enzymes, $r_{\text{inv}}$, were calculated across different combinations of physiologically relevant sucrose and hexose concentrations to determine the metabolic function of sucrose, that is, $f(\text{Suc}) = r_{\text{in}} - r_{\text{inv}}$. Simulation results of different sucrose concentrations were plotted against glucose and fructose concentrations (Figure 1). Thus, each shown plane in the figure corresponds to solutions of $f(\text{Suc})$, $J$ and $H$ for one sucrose concentration (a detailed definition of concentrations is provided in the figure legend). Although sucrose concentrations used for 4°C simulations were up to 8-fold higher than under 22°C, resulting absolute values and dynamics of $f(\text{Suc})$ were significantly lower than under 22°C (Figure 1a,b). Reduced absolute values were due to a decreased input rate $r_{\text{in,4°C}}$ (based on experimental findings). As expected, under conditions of low product concentration, $f(\text{Suc})$ became minimal under both temperatures due to increased rates of sucrose cleavage (Figure 1a,b). However, reduced dynamics of $f(\text{Suc})$ was due to increased hexose concentrations (inhibitors) and a reduced

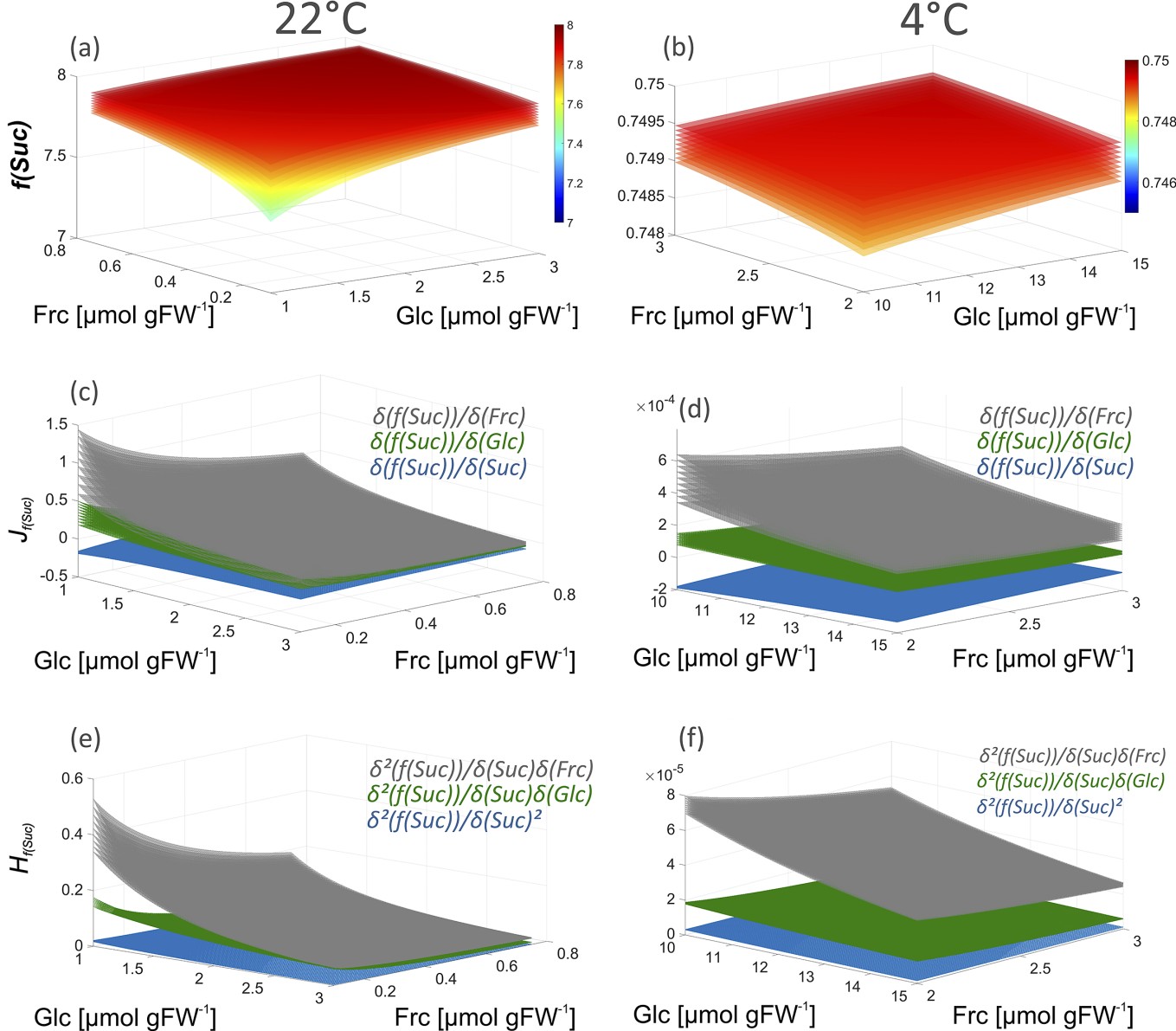

**Fig. 1.** Dynamics of $f(Suc)$ under ambient and low temperature. (a) $f(Suc)$ at 22°C under variable concentrations of fructose ($x$-axis), glucose ($y$-axis) and sucrose (planes). (b) $f(Suc)$ at 4°C under variable concentrations of fructose ($x$-axis), glucose ($y$-axis) and sucrose (planes). Unit of $f(Suc)$: ($\mu$mol Suc h$^{-1}$ gFW$^{-1}$). (c) Jacobian matrix entries of $f(Suc)$ at 22°C under variable concentrations of glucose ($x$-axis), fructose ($y$-axis) and sucrose (planes; see equation S5 in the supplements; $j_{11}$: blue; $j_{12}$: green; $j_{13}$: grey). (d) Jacobian matrix entries of $f(Suc)$ at 4°C under variable concentrations of glucose ($x$-axis), fructose ($y$-axis) and sucrose (planes). See equation S5 in the supplements; $J_{11}$: blue; $J_{12}$: green; $J_{13}$: grey. (e) Hessian matrix entries of $f(Suc)$ at 22°C under variable concentrations of glucose ($x$-axis), fructose ($y$-axis) and sucrose (planes), see equation S11 in the supplements; $h_{f(Suc,11)}$: blue; $h_{f(Suc,12)}$: green; $h_{f(Suc,13)}$: grey. (f) Hessian matrix entries of $f(Suc)$ at 4°C under variable concentrations of glucose ($x$-axis), fructose ($y$-axis) and sucrose (planes), see equation S11 in the supplements; $h_{f(Suc,11)}$: blue; $h_{f(Suc,12)}$: green; $h_{f(Suc,13)}$: grey. Each plane corresponds to a sucrose concentration which was varied between 1–3 $\mu$mol gFW$^{-1}$ and 4–8 $\mu$mol gFW$^{-1}$ for simulations at 22 and 4°C, respectively.

$V_{max}$ of invertase (see Table 1). As a result, also the dynamic range of $J$ and $H$ decreased across all simulated scenarios by several orders of magnitude ($10^{-1} \rightarrow 10^{-4}/10^{-5}$; Figure 1c–f). A main low temperature effect became visible in entries of Jacobian matrices which was a reduced degree of overlap between $j_{12}\left(\frac{\partial(f(Suc))}{\partial(Glc)}\right)$ and $j_{13}\left(\frac{\partial(f(Suc))}{\partial(Frc)}\right)$ (Figure 1c,d). Both terms describe changes of $f(Suc)$ induced by (slight) changes of glucose and fructose concentrations, respectively. At 22°C, high glucose concentrations (~ 2.5–3 $\mu$mol gFW$^{-1}$) minimise $j_{13}$ and, with this, also the regulatory effect of fructose dynamics on $f(Suc)$ (see Figure 1c). At 4°C, high glucose

concentrations (~ 14–15 $\mu$mol gFW$^{-1}$) also lead to minimal values of $j_{13}$, which were, however, still significantly higher than $j_{12}$ (see Figure 1d; ANOVA, $p < .001$). This discrepancy became also visible in the curvature of $f(Suc)$, that is, in the Hessian matrix (Figure 1e,f).

These observations suggest that, under ambient conditions and (increased) glucose concentrations, it is $j_{12} \approx j_{13}$, that is, $\frac{\partial(f(Suc))}{\partial(Glc)} \approx \frac{\partial(f(Suc))}{\partial(Frc)}$, and $h_{f(Suc),12} \approx h_{f(Suc),13}$, that is, $\frac{\partial^2(r_{in}-r_{inv})}{\partial(Suc)\partial(Glc)} \approx \frac{\partial^2(r_{in}-r_{inv})}{\partial(Suc)\partial(Frc)}$. At low temperature, this similarity is not given even under (relatively) high glucose concentrations

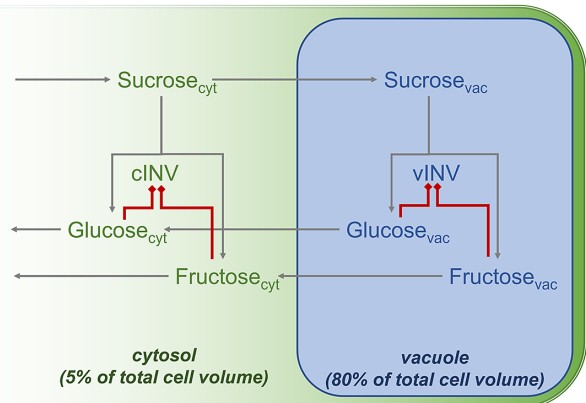

**Fig. 2.** Schematic overview of subcellular invertase reactions. Green colour indicates cytosolic metabolites and enzymes and blue colour indicates vacuolar metabolites and enzymes. For simulation of subcellular sucrose cleavage, effective metabolite concentrations were calculated based on assumptions and findings of previous studies (Kitashova et al., 2021; Nägele & Heyer, 2013). Based on previous findings (Fürtauer et al., 2016), subcellular sugar distribution was assumed as follows: cytosolic sucrose, 22°C: 50%; vacuolar sucrose, 22°C: 25%; cytosolic hexoses, 22°C: 30%; vacuolar hexoses, 22°C: 55%; cytosolic sucrose, 4°C: 40%; vacuolar sucrose, 4°C: 33%; cytosolic hexoses, 4°C: 30%; vacuolar hexoses, 4°C: 50%.

which might suggest a cold-induced switch of the regulatory role which fructose plays in plant metabolism (Klotke et al., 2004).

## 3. Vacuolar metabolite transport increases elasticity of the cellular sucrose function

To study a regulatory effect of hexose accumulation under low temperature in more detail, the model was extended to subcellular distribution of sugars and invertase isoforms (Figure 2). Effective cytosolic and vacuolar sugar concentrations in *Arabidopsis* leaf mesophyll cells were estimated as previously described assuming the cytosol to comprise 5% and the vacuole 80% of the total cell volume (Nägele & Heyer, 2013). Further, subcellular sugar distribution at 22 and 4°C, respectively, was derived from a previous study (Fürtauer et al., 2016). Details about the relative distribution of sugars are provided below (see legend of Figure 2).

Assuming a volume of 1 ml $H_2O$ to equal (approximately) 1 g fresh weight of *Arabidopsis* leaf material (Nägele & Heyer, 2013), effective sugar concentrations were derived from the sugar amounts used before (see Figure 1; Kitashova et al., 2021). To simulate compartment-specific sucrose cleavage, neutral (cytosolic) and acidic (vacuolar) invertase activities were considered separately (Kitashova et al., 2021). Subcellular simulations of $f(\text{Suc})$ across a physiologically feasible range of metabolite concentrations at 22°C revealed higher variability of vacuolar metabolic functions than in the cytosol (Figure 3a,d). Entries of Jacobian and Hessian matrices, which accounted for changes in substrate and/or product concentrations, differed by orders of magnitude between cytosolic and vacuolar reactions (Figure 3b,c,e,f).

These differences were due to 16-fold dilution of metabolites comparing the vacuolar and cytosolic volume (5 vs. 80% of the total cell volume). Hence, lowered effective metabolite concentrations in the vacuole resulted in higher elasticity of $f(\text{Suc})$ due to lowered invertase inhibition by glucose and fructose. At 4°C, the discrepancy between concentration effects on $f(\text{Suc})$ in cytosol and vacuole became stronger due to significant cold-induced sugar accumulation (Figure 3g,j). Under these conditions, $f(\text{Suc})$ in the cytosol was almost invariant across the simulated sucrose concentration range of 25.6–51.2 mM (Figure 3g) while dynamics were still observable for the vacuolar $f(\text{Suc})$ (Figure 3j). This was numerically reflected in Jacobian and Hessian matrix entries of the

subcellular metabolic function of $f(\text{Suc})$ which revealed dynamics of vacuolar fructose concentration to have the strongest regulatory effect within the simulated scenario (Figure 3h,i,k,l).

Together with the findings of the whole cell model (Figure 1), these observations suggest that transport of sucrose into the vacuole maximise effects of metabolic regulation on $f(\text{Suc})$ and provide further evidence for a dominant role of fructose in regulation of sucrose cleavage under low temperature. Previous studies have shown that sugar accumulation, in general, plays a central role in plant cold response and acclimation (Guy et al., 2008; Hannah et al., 2006; Seydel, Kitashova, et al., 2022). Fructose and its phosphorylation product, fructose 6-phosphate (F6P), have been found to significantly contribute to stabilisation of a plant metabolic homeostasis during cold exposure (Bogdanović et al., 2008). F6P is a direct product of the Calvin Benson Cycle and serves as substrate for many other metabolic pathways, for example, starch biosynthesis, sucrose biosynthesis and glycolysis (Ruan, 2014). Thus, findings of the present study suggest that tight regulation of $f(\text{Suc})$ in the cytosol and vacuole by fructose directly connects sucrose dynamics with the stabilisation of many other cellular pathways. In future studies, a combination of the presented kinetic approach with subcellular sugar analysis of mutants being affected in sucrose cleavage and subcellular sugar transport might reveal further detailed insights into the regulatory network of plant sucrose metabolism.

## 4. Conclusions

Together with the Jacobian matrix, Hessian matrices are commonly applied to study *n*-dimensional functions and surfaces, their extrema and their curvature (see e.g., (Basterrechea & Dacorogna, 2014; Ivochkina & Filimonenkova, 2019). In context of the presented theory for analysis of biochemical metabolic functions, this suggests that metabolism can be summarised by a multi-dimensional function which supports the analysis of complex metabolic regulation, for example, of metabolic cycling. Although calculation of metabolic functions, Jacobian and Hessian matrices is straight forward, it essentially supports quantitative analysis of multi-dimensional dynamics, shape and curvature of a metabolic landscape (Figure 4).

Findings of the present study emphasise the necessity to resolve eukaryotic metabolism to a subcellular level in order to reliably

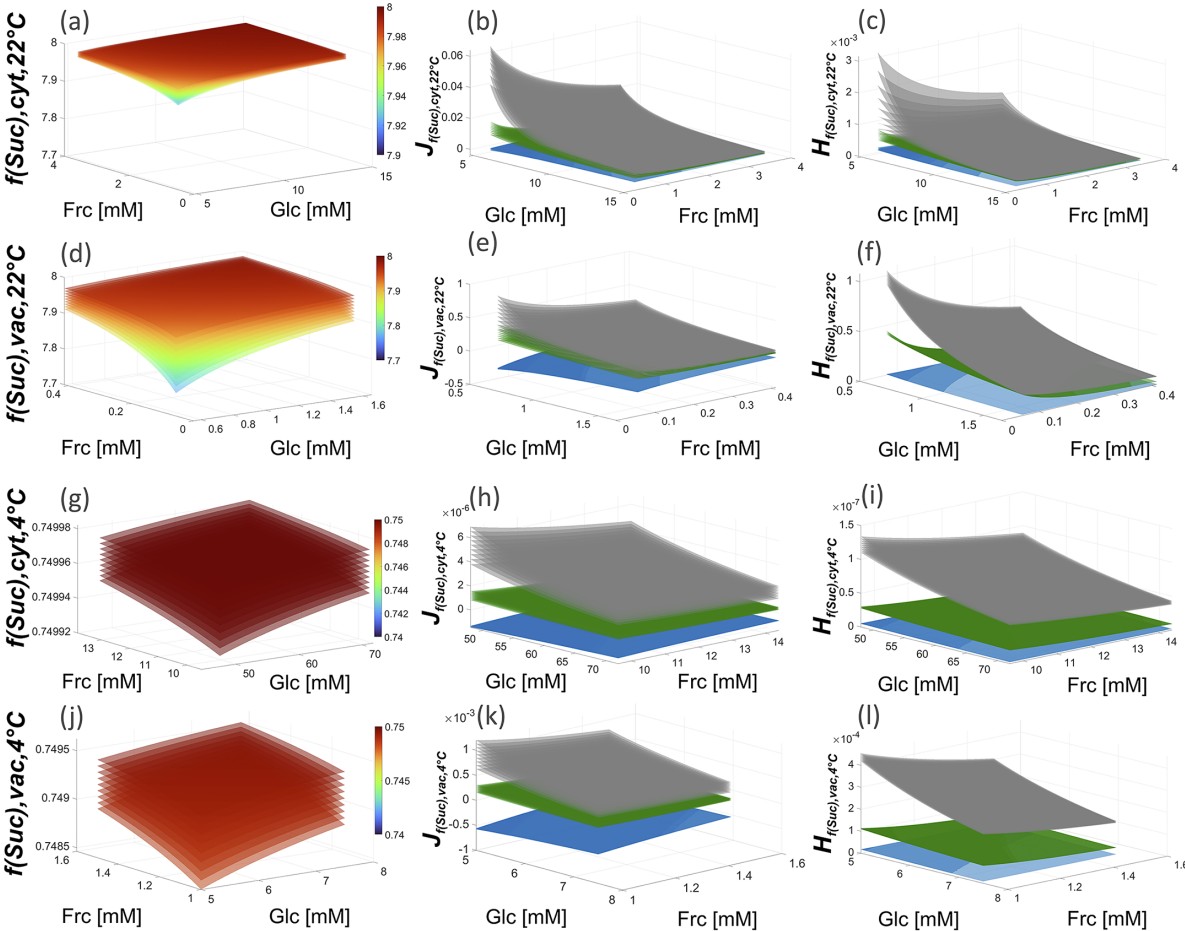

**Fig. 3.** Estimated cytosolic and vacuolar dynamics of $f$(Suc) under ambient and low temperature at variable concentrations of fructose, glucose and sucrose. Planes represents simulations for different sucrose concentrations (sucrose concentration ranges: cytosol, 22°C: 8–24 mM; vacuole, 22°C: 0.25–0.75 mM; cytosol, 4°C: 25.6–51.2 mM; vacuole, 4°C: 1.32–2.64 mM). (a–f) estimations at 22°C, (g–l) estimations at 4°C. (a) cytosolic $f$(Suc) at 22°C, (b) Jacobian entries $j_{11}$ (blue), $j_{12}$ (green), $j_{13}$ (grey) of cytosolic $f$(Suc) at 22°C, (c) Hessian entries $h_{f(Suc,11)}$ (blue), $h_{f(Suc,12)}$ (green), $h_{f(Suc,13)}$ (grey) of cytosolic $f$(Suc) at 22°C, (d) vacuolar $f$(Suc) at 22°C, (e) Jacobian entries $j_{11}$ (blue), $j_{12}$ (green), $j_{13}$ (grey) of vacuolar $f$(Suc) at 22°C, (f) Hessian entries $h_{f(Suc,11)}$ (blue), $h_{f(Suc,12)}$ (green), $h_{f(Suc,13)}$ (grey) of vacuolar $f$(Suc) at 22°C, (g) cytosolic $f$(Suc) at 4°C, (h) Jacobian entries $j_{11}$ (blue), $j_{12}$ (green), $j_{13}$ (grey) of cytosolic $f$(Suc) at 4°C, (i) Hessian entries $h_{f(Suc,11)}$ (blue), $h_{f(Suc,12)}$ (green), $h_{f(Suc,13)}$ (grey) of cytosolic $f$(Suc) at 4°C, (j) vacuolar $f$(Suc) at 4°C, (k) Jacobian entries $j_{11}$ (blue), $j_{12}$ (green), $j_{13}$ (grey) of vacuolar $f$(Suc) at 4°C, (l) Hessian entries $h_{f(Suc,11)}$ (blue), $h_{f(Suc,12)}$ (green), $h_{f(Suc,13)}$ (grey) of vacuolar $f$(Suc) at 4°C. Colour bars in the left panel (a,d,g,j) indicate values of $f$(Suc).

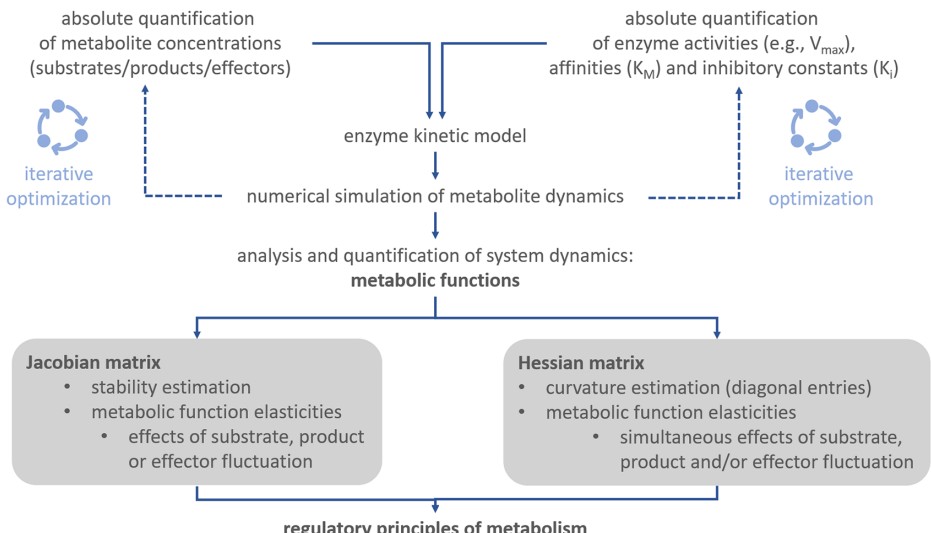

**Fig. 4.** Workflow for deriving regulatory principles of metabolism.

estimate dynamics of metabolite concentrations in terms of reaction rates and transport processes. Finally, applying such analysis to dynamic metabolic systems can unravel non-intuitive regulatory patterns. This supports the quantitative interpretation of experimental observations on metabolism within a dynamic environment.

## Acknowledgements

I would like to thank all members of Plant Evolutionary Cell Biology, LMU Munich, for many fruitful seminars and discussions. Special thanks go to Lisa Fürtauer, RWTH Aachen, and AG Weckwerth, University of Vienna, as well as Jakob Weiszmann and Matthias Nagler for constructive advice, support and discussion. Finally, I thank the SFB/TR175 consortium for a supportive research environment and fruitful discussions.

**Financial support.** This work was supported by Deutsche Forschungsgemeinschaft (DFG), grants TR175/D03 and NA 1545/4-1.

**Conflict of interest.** The author declares no conflicts of interest.

**Data availability statement.** The main data supporting the findings of this study are contained within the article and cited literature.

**Supplementary Materials.** To view supplementary material for this article, please visit http://doi.org/10.1017/qpb.2022.5.

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
