## [Reviewer Report]

Dear Dr. Hamant,

Dear Dr. Fleck,

I am submitting a theories article to be considered for publication in your journal Quantitative Plant Biology. The article is entitled “Metabolic regulation inferred from Jacobian and Hessian matrices of metabolic functions”. It presents a theoretical approach to develop first-order and second-order partial derivatives of metabolic functions summarised by Jacobian and Hessian matrices. Applying a simple kinetic model of invertase-driven sucrose hydrolysis, an approach was developed for quantitative analysis of kinetic regulation of sucrose metabolism based on partial derivatives of metabolic functions. Based on previously published experimental observations, metabolite dynamics were quantitatively explained in context of underlying metabolic functions and enzyme kinetics. I applied this approach to explain differential regulation of sucrose and hexose concentrations in Arabidopsis thaliana during cold acclimation. Finally, a switch of the metabolic role of fructose metabolism is suggested to be involved in metabolic reprogramming during plant cold acclimation. The manuscript has been made publicly available on the bioRxiv pre-print server (https://doi.org/10.1101/2021.10.05.463227).

In summary, I am convinced that this article is of high interest for the readership of Quantitative Plant Biology. I would be pleased to publish it within your journal and thank you in advance for editing my manuscript.

Sincerely yours,

Thomas Nägele

---

## [Reviewer Report]

*Comments to Author*: Prof. Nägele presents a theoretical study in which he applies the Jacobian and Hessian matrix of metabolic functions to analyse a simple model of invertase-driven sucrose hydrolysis. Using a recently published set of carbon uptake rates and kinetic parameters from Arabidopsis thaliana, accession Col-0 at ambient temperature and cold stress the author studies the potential regulatory role of metabolite dynamics during plant cold acclimation. The manuscript is well written and sound. However, as the concept of applying partial derivatives to study metabolite dynamics is not novel, the manuscript is currently lacking convincing insights that can be gained from the presented analysis. Thus, I have a few major and minor suggestions, most importantly:

The manuscript would gain matter if the author elaborated more on the biological insights of this computational analysis and how sucrose hydrolysis affects metabolism under normal conditions and cold stress.

Minor points:

The term “metabolic function” can have different meanings in different scientific communities and contexts. Thus, defining this term even more explicitly, both in the abstract and the introduction, could avoid confusing the reader.

It would be insightful for the reader if this analysis was placed in a larger context and to learn in which other studies the Jacobian and/or Hessian matrix has been used to study metabolite dynamics.

Although the manuscript is well written and enjoyable for an expert reader it might be challenging to follow for a non-expert reader. However, I guess that might be expected from a manuscript to be published in Quantitative Plant Biology.

---

## [Reviewer Report]

*Comments to Author*: The paper of Nägele entitled “Metabolic regulation inferred from Jacobian and Hessian matrices of metabolic functions” introduces a mathematical modeling framework based on the estimation of Jacobian and Hessian matrices of a metabolic network at a steady state to describe influences of metabolites and conditions on fluxes and metabolite concentrations. It is applied to a simple model of invertase-driven sucrose hydrolysis using literature data under two different temperatures.

The interdisciplinary combination of experiments and systems theoretic analysis is generally interesting and promising. However, while reading the paper, I had several difficulties in understanding the overall aim of the study and to follow the line of argumentation. I was not able to extract the main research results. I suggest to re-structure the paper in this respect and hope that my detailed comments below help to do so. Moreover, I recommend considering teaming up with a partner from the mathematical/modeling field, to clarify the issues on the mathematical side.

Major points:

I found the paper difficult to read, mainly due to the structure and line of argumentation:

- The introduction contains a short explanation the complexity of enzyme activities in cellular systems, explains the well known law of mass action and states that metabolic systems can be described by differential equations.

In an introduction, I would rather expect that the intro describes current state of work in the field, their limitations and open research questions / problems, and then states how (some of) these are addressed in the paper and which methods are used, and probably a short outlook on the main results of the paper.

- The section “Deriving a Jacobian matrix to study ….” (p4ff) still reads as a kind of introduction / review for a derivation of the MM kinetics. When the sucrose hydrolysis system was introduced as a model (Eq4), it was not clear to me that this is the particular application system for the paper. In the following, product inhibition of invertases are described (which means that hexoses inhibit r_inv in the model, if I am not mistaken). Since this is crucial also for the following, I suggest to improve the reaction scheme to make this dependence clear, and probably cast it into a Figure, which illustrates the example system that is analyzed here. In connection with the mathematical model (Eq 8-11).

- I also suggest to design a figure that illustrates the methodological pipeline. Key messages do not become clear from only one Figure in the end of the paper.

- p6: are equations 8,9 extracted from other literature sources? Please clearly indicate

-p7, first line: Now that the model is introduced, one expects the description of questions and goals in the context of this model latest; instead, a description of linearization to study stability of a steady state follows. I do not understand the message of this paragraph. The model introduced on p6 clearly reaches a dynamic steady state with constant in- and effluxes, if I am not mistaken. The time scale of this convergence of course depends on initial concentrations and parameter values for rate constants and Michaelis constants. Now it is stated that plant metabolism can hardly be described as in steady state due to its external and internal dynamics, which is, however, not included in the model. The motivation why the analysis via Jacobian and Hessian matrices makes sense at all if the steady state assumption is not fulfilled is not clear to me. Moreover, the assumption that the velocity of an enzymatic reaction linearly depends on substrate concentration is in contrast to the before mentioned MM kinetics, where the order of the reaction velocity with respect to the substrate changes from one to zero with increasing substrate concentration.

- It is not completely clear what the Jacobian matrix can be used for. It is argued that the spectrum of J_f indicates stability of a steady state, but is stability really a relevant issue for the particular system at hand? Without knowing the Jacobian matrix and the exact dynamics, I would intuitively say that this is a stable system with respect to perturbations of any concentrations of species in the network in a steady state.

- A critical discussion of limitations of the method / results is missing. How do results relate to other similar studies? Is this approach e.g. related to Modular response analysis, which is to my knowledge derived from Metabolic Control Analysis?

- The application in connection with the data gathered in Table 1 is interesting, but the description is not sufficient to understand the key points: what is the goal of this application / what are the questions, how are r_inv rates exactly calculated, what do we learn from the results?

Minor

- p3: explanation mass action law: “This probability is proportional to (I) the concentration of reactants, (ii) to the number of molecules which participate in a reaction”: this is a misleading wording. The number of molecules usually appears in stochastic descriptions, where molecule numbers are low. Otherwise, concentrations are used in the law of mass action.

- p3, Equation 1, similarly: what are S_i and P_j, are these concentrations or molecule numbers?

-p7, Jacobian matrix: I would rather explain linearization by the ODE system for the dynamics of a small perturbation from a steady state, i.e. \dot (\Delta x)=J_f(\bar x)\Delta x, where \Delta x=x-\bar x is the deviation from the steady state \bar x and J_f(\bar x) is the Jacobian matrix of the ODE system evaluated at the steady state (which is important but missing in the text).

- p8, sentence: Further, these equations show:… is this clear from the equations? I think this is generally intuitively clear when using MM kinetics to describe substrate conversions

- p8: what is a “square decrease towards zero”?

-p8, equ. 16 not clear to me. If hexose is present in large amounts, this inhibits r_inv and, if r_in is constant, this leads to an accumulation of Suc, right? Can’t this be seen just by the reaction scheme with known feedback influences?

---

## [Reviewer Report]

*Comments to Author*: Dear Prof. Nägele,

we have now received the required number of review reports. Unfortunately, both reports are critical and request major rewriting of your manuscript. 

I suggest that you take into account the comments of the reviewers and revise the manuscript accordingly. In its current form the manuscript is not appropriate for publication in Quantitative Plant Biology. 

Kind regards,

Christian Fleck

---

## [Reviewer Report]

Dear Dr. Hamant,

Dear Dr. Fleck,

I am submitting a fully revised theories article to be considered for publication in your journal Quantitative Plant Biology. Based on reviewer comments, the article is now entitled “Metabolic regulation of subcellular sucrose cleavage inferred from Jacobian and Hessian matrices of metabolic functions”. I have addressed all comments raised by both reviewers and have extended my theoretical analysis to a subcellular level of metabolism. My findings provide strong evidence for an important role of vacuolar sucrose transport during cold acclimation. To my opinion, this provides strong evidence for the applicability and usefulness of my approach. I have further added a methodological pipeline graph to indicate how metabolic functions and their dynamics might be numerically analysed to reveal detailed insights into metabolic regulation. To facilitate further review activities, I have highlighted all changes made (yellow background).

In summary, I am convinced that this revision has improved article is of high interest for the readership of Quantitative Plant Biology. I would be pleased to publish it within your journal and thank you in advance for editing my manuscript.

Sincerely yours,

Thomas Nägele

---

## [Reviewer Report]

*Comments to Author*: The author has addressed all my suggestions and questions in a satisfying manner. The revised version of the manuscript has much gained from the addition of more biological context. I only have two minor comments left.

Axis labels of Figure 1 and especially Figure 3 are fairly small and could be increased for better readability.

Figure 2 seems to be corrupted during the submission process.

---

## [Reviewer Report]

*Comments to Author*: I apologise for the considerable time it took to review the manuscript, but we had difficulties to find the needed number of reviews. One reviewer decided to stay anonymous. I therefore attach the review below. 

I recommend to carefully consider the comments of this reviewer. In particular, I would like to remind the author that a research article is not a review article. I suggest to shorten the introductory material as also proposed by the reviewer. 

Report from anonymous reviewer:

In my opinion, the revised manuscript has improved compared to the original one. In particular, some of the newly added paragraphs about the interpretation of the findings of the study in the biological context are helpful. However, I still think that the manuscript needs some major revisions in order to make it accessible to a broad readership.

My major concern is about the structure of the paper. While this has partly improved in the revised version, I still think that it contains too much text about „standard knowledge“ (Michaelis Meinten kinetics, definition and calculation of Jacobian and Hessian matrices…). It takes the reader a long time until the first results are presented. Moreover, at this point the „study design“ (i.e. which questions are to be answered, with which methods and why) is still not completely clear. I could also not find a concise summary of the major findings of the paper. 

To my knowledge, the theory about metabolic networks is vast, and interpretations of Jacobian (and maybe also Hessian?) matrices can also be found elsewhere. So I suggest to re-structure the content of the paper and to 1. shorten standard knowledge considerably, 2. think about how to even more emphasize and illustrate the major findings, 3. refer to existing literature about using J and H for an analysis of metabolic networks.

It is also still not completely clear to me what is to be estimated. The Jacobian and Hessian matrix of a steady state are just matrixes with fixed entries that can readily be calculated if all model parameters are given. And if I understand correctly, this is what is done to produce the results. However, the author elaborates on estimating Jacobian matrices from experimental data, which is a much harder problem. I do not see the connection to his results. 

The added paragraph on p7 about why a steady state assumption is justified is helpful, but should be integrated earlier in the manuscript.

---

## [Reviewer Report]

Dear Dr. Hamant,

Dear Dr. Fleck,

based on your and reviewer comments, I have revised my article entitled “Metabolic regulation of subcellular sucrose cleavage inferred from quantitative analysis of metabolic functions”. I have addressed all comments raised by you and both reviewers. I have added a supplemental file to explain the mathematical basics behind the analysis to prevent having too much text in the main manuscript/introduction.

I hope I can convince you that this revision has improved the article and can be considered for publication in Quantitative Plant Biology. 

Thank you very much for editing my manuscript!

Sincerely yours,

Thomas Nägele

---

## [Reviewer Report]

*Comments to Author*: Dear Dr. Nägele,

I find your manuscript now sufficiently improved and ready for publication in QPB. 

Kind regards,

Christian Fleck